# Microbiological Quality of Deer Meat Treated with Essential Oil *Litsea cubeba*

**DOI:** 10.3390/ani12182315

**Published:** 2022-09-06

**Authors:** Simona Kunová, Esther Sendra, Peter Haščík, Nenad L. Vuković, Milena D. Vukić, Anis Ben Hsouna, Wissem Mnif, Miroslava Kačániová

**Affiliations:** 1Institute of Food Technology, Faculty of Biotechnology and Food Sciences, Slovak University of Agriculture, Trieda A. Hlinku 2, 94976 Nitra, Slovakia; 2Centro de Investigación e Innovación Agroalimentaria y Agroambiental (CIAGRO-UMH), Escuela Politécnica Superior de Orihuela, Miguel Hernández University, 03312 Orihuela, Spain; 3Department of Chemistry, Faculty of Science, University of Kragujevac, R. Domanovica 12, 34000 Kragujevac, Serbia; 4Laboratory of Biotechnology and Plant Improvement, Centre of Biotechnology of Sfax, Sfax 3038, Tunisia; 5Department of Life Sciences, Faculty of Sciences of Gafsa, Zarroug, Gafsa 2112, Tunisia; 6Department of Chemistry, Faculty of Sciences and Arts in Balgarn, University of Bisha, P.O. Box 199, Bisha 61922, Saudi Arabia; 7ISBST, BVBGR-LR11ES31, Biotechpole Sidi Thabet, University of Manouba, Ariana 2020, Tunisia; 8Institute of Horticulture, Faculty of Horticulture and Landscape Engineering, Slovak University of Agriculture, Trieda A. Hlinku 2, 94976 Nitra, Slovakia; 9Department of Bioenergetics, Food Analysis and Microbiology, Institute of Food Technology and Nutrition, Rzeszow University, Cwiklinskiej 1, 35-601 Rzeszow, Poland

**Keywords:** deer meat, essential oils, MALDI-TOF, meat safety, vacuum-packaging

## Abstract

**Simple Summary:**

Consumers are increasingly turning to healthier and less environmentally harmful diet alternatives. Game is an ideal food from this point of view because it represents meat with a high protein content, low fat content, a favourable composition of fatty acids and minerals. Various types of packaging are often used to extend the shelf life of meats. Packaging can be combined with natural antimicrobials, such as various plant extracts and essential oils, for better effectiveness. Little is known about the microbial quality and preservation of deer meat. In the present study, deer meat was treated with essential oil from *Litsea cubeba* 0.5 and 1.0% concentration in rapeseed oil combined with aerobic and vacuum packaging. The meat was evaluated for microbiological quality (counts and microbiota identification) for 20 days under refrigerated storage. Our result show that *Litsea cubeba* essential oil is an effective natural agent against deer meat spoilage bacteria.

**Abstract:**

The present study aimed to evaluate deer meat microbiological quality when treated with essential oil (EO) from *Litsea cubeba* (dissolved in rapeseed oil at concentrations 0.5 and 1%), in combination with vacuum packaging during 20 days of storage of meat at 4 °C. Total viable counts (TVC), coliforms bacteria (CB), lactic acid bacteria (LAB) and *Pseudomonas* spp. were analysed at day 0, 1, 5, 10, 15 and 20. MALDI-TOF MS Biotyper technology was applied to identify microorganisms isolated from meat. The highest number of TVC at the end of the experiment was 5.50 log CFU/g in the aerobically packaged control group and the lowest number of TVC was 5.17 log CFU/g in the samples treated with 1.0% *Litsea cubeba* EO. CB were not detected in the samples treated with 1.0% *Litsea cubeba* EO during the entire storage period. Bacteria of the genus *Pseudomonas* were detected only in the aerobically and vacuum packaged control group. The highest number of LAB was 2.06 log CFU/g in the aerobic control group, and the lowest number of LAB was 2.01 log CFU/g in the samples treated with 1.0% *Litsea cubeba* EO on day 20. The most frequently isolated bacteria from deer meat were *Pseudomonas ludensis*, *Pseudomonas corrugata*, *Pseudomonas fragi*, *Bacillus cereus*, *Staphylococcus epidermidis* and *Sphingomonas leidyi*.

## 1. Introduction

The quality of wild animal meat is influenced by various external and internal factors. The external ones include climatic and seasonal conditions, which affect pastures, as well as the environment in which the animals live, the internal factors are mainly age, sex, physical and sexual activity of the animals [1]. Meat is a very good source of bioactive substances that are essential for human nutrition. Deer meat is a very valuable source of vitamins A, C, E and B, as well as minerals Fe, K, Ca, Mg, Cu, Zn and Se. Deer meat is rich in calcium, fluorine, iron, copper, zinc and chromium. The strong aroma and sweet taste of deer meat combined with the metallic flavour is created by the high iron content in the meat [2]. Game meat is low in fat and has a high protein content [3], nowadays it is gaining popularity as it is perceived as healthy, as well as linked to regional/traditional diet [4]. Obtaining meat from wild game is significantly different from obtaining meat from livestock. Deers are hunted, and death is caused by a gunshot (usually in the head, neck or chest), in such process, the ante mortem inspection (before shooting) is performed by the hunter, and the post-mortem inspection (after shooting and dissection) is performed either by the hunter with specific training or by a veterinarian [5]. The microbial quality of meat from hunted animals is affected by several factors, the most common of which are: poor wound placement, muscle contamination by gastrointestinal discharge or faeces during expulsion, and delayed or insufficient cooling [6]. Bacteria commonly found in chilled meat are *Pseudomonas* spp., *Lactobacillus* spp. and Enterobacteriaceae. *Pseudomonas* spp. is dominant in spoiled meat. Facultative anaerobic bacteria, genus *Lactobacillus* spp. dominate in vacuum-packaged meat [7]. The spoiling microflora of vacuum-packaged meat is represented mainly by lactic acid bacteria. They are able to grow in the presence but also in the absence of oxygen, considerably limiting storability under anaerobic conditions [8]. Bacteria of the genus *Pseudomonas* have been identified as responsible for the deterioration of fresh meat stored under aerobic conditions. They are among the fastest growing organisms in aerobic conditions at low temperatures [9]. Coliforms bacteria are found in water, soil, grains, blooming flowers and trees, they are found on fruits, vegetables but also on insects, animals and humans. They are commonly used as indicators of the sanitary quality of food and water [10]. Little is known about the microbial populations in meat from hunted animals, in-depth studies are needed to provide knowledge in this field. MALDI-TOF MS Biotyper is a fast and reliable method of identification of the microbial isolates. This Biotyper has several unique features, making it one of the most successful identification methods [11]. MALDI-TOF MS is applicable in various industries, whether in the clinical, environmental or food spheres. It can be used to assess the quality and safety of food and to verify the authenticity of food [12] and would be a valuable technology to investigate microbial populations in meat.

The identification of the microbial populations in meat would be a first step towards defining strategies for meat preservation. The evaluation of meat storage conditions, such as packaging conditions combined with the use of natural antimicrobial substances, would be another contribution to meat safety. In this sense, some plant extracts, mainly essential oils, have antimicrobial properties that can be potentially suitable for meat preservation. *Litsea cubeba* (LC) or Cuban laurel is a small dioecious deciduous tree or shrub that occurs wild in tropical and subtropical regions, especially in Asia, Malaysia, Indonesia, China, and Taiwan [13]. Plants produce secondary metabolites, which include essential oils (EO), that help in defending themselves against various pests, such as insects, as well as various fungi. *Litsea cubeba* essential oils are characterised by antibacterial, antiparasitic, antifungal, antioxidant, antiseptic, antiviral and insecticidal effects and have potential uses for different industries [14,15]. Essential oils that contain high concentrations of phenolic compounds, such as carvacrol, eugenol and thymol, have the strongest antimicrobial effects [16], however many other compounds provide valuable antimicrobial properties [16]. All parts of the *Litsea cubeba* plant contain essential oil (EO). EO content in the fruit is in the range of 0.3–5% [17,18]. EO from *Litsea cubeba* has a light-yellow colour, a fresh sweet-fruity lemon scent. It is used in the chemical, cosmetic and medical industries, and, it is also added to food and tobacco products. The main chemical component of the EO from *Litsea cubeba* is citral, which is a mixture of stereoisomers geranial and neral. Other ingredients include limonene, methylheptane and pinene to linalool. However, the chemical composition is variable, depending on the cultivation place and climate [19]. The interaction between the main and secondary components of the oil, which have a synergic effect on each other, is mainly responsible for the inhibitory activity of the essential oil [20]. At present, some information is available on the in vitro antimicrobial activity of LCEO against food spoilage bacteria [21], however little information is available regarding studies in real food systems. To our knowledge, this is the first study on the use of LCEO in meat preservation, and specifically in game meat.

The present study was aimed to evaluate the microbiological quality of vacuum packaged deer meat stored at 4 °C for 20 days and treated with 0.5 and 1.0% essential oil from *Litsea cubeba*.

## 2. Materials and Methods

### 2.1. Preparation and Packaging of Deer Meat Samples

Thigh muscle samples of deer meat (*musculus semimembranosus*) were used in this experiment. The meat was bought from an authorised store; the meat sample was obtained from a young male deer living in the wild, according to label information estimated age was 2 years, and it was bought 48 h after the catch. The animal was shot by a bullet in the neck area, in the village of Devičany (48.3236° N, 18.7074° E), Nitra region, in Slovakia. The meat samples were transported under hygienic conditions in a cleaned refrigerator to the microbiological laboratory, where they were stored at temperature 4 °C until the analysis was performed. Samples were transported from the authorised store to the laboratory within 30 min. Meat was diced, samples weighing 5 g were treated with solutions of 0.5 and 1.0% *Litsea cubeba* fruit essential oil (LCEO) (Hanus, Nitra, Slovakia) dissolved in rapeseed oil and vacuum packaged using a vacuum packer (Concept, Choceň, Czech Republic). Food grade rapeseed oil was purchased from an authorised store. The EO was previously characterised by Borotová et al. [21]. A total of 120 meat samples were analysed. The samples were prepared as follows:Control aerobically packaged group: Meat samples were packed in polyethylene bags under aerobic conditions and stored at 4 °C;Control group with vacuum packaging: Samples of fresh meat were packed in polyethylene bags and stored under anaerobic conditions at 4 °C;Control group with rapeseed oil: Meat was treated with rapeseed oil, packaged in polyethylene bags, and stored under anaerobic conditions at 4 °C;Vacuum-packaged with 0.5% *Litsea cubeba* essential oil: Meat samples were soaked in a solution of rapeseed oil containing 0.5% *Litsea cubeba* essential oil, then packed and vacuum sealed in polyethylene bags, and further stored under anaerobic conditions at 4 °C;Vacuum-packaged treated with 1.0% *Litsea cubeba* essential oil: Meat samples were soaked in a solution of rapeseed oil containing 1.0% *Litsea cubeba* essential oil, packed and vacuum sealed in polyethylene bags, and stored under anaerobic conditions at 4 °C.

For the application of the EO, meat samples were soaked in the solution of *Litsea cubeba* essential oil for 30 min.

### 2.2. Samples Cultivation

Microbiological analyses were performed on days 0, 1st, 5th, 10th, 15th and 20th of storage at 4 °C. Five gram samples were diluted with 45 mL of 0.1% sterile saline solution. The samples were homogenised in a shaker (GFL 3031, Burgwedel, Germany) for 30 min. The following microbial populations were evaluated: Lactic acid bacteria (LAB) were determined in media De Man, Rogosa and Sharpe agar (MRS, Oxoid, Basingstoke, UK) incubated with 5% of CO_2_ at 37 °C for 48–72 h. *Pseudomonas* were determined using Pseudomonas agar (Oxoid, Basingstoke, UK) incubated at 35 °C for 48 h. Coliforms bacteria were determined in Violet Red Bile Lactose Agar (VRBL, Oxoid, Basingstoke, UK) incubated at 37 °C for 24 to 48 h. Total viable counts were determined in Plate Count Agar (PCA, Oxoid, Basingstoke, UK) incubated at 30 °C for 48–72 h.

### 2.3. Identification of Microorganisms by MALDI-TOF MS

MALDI-TOF (matrix-assisted laser desorption/ionisation time of flight) MS Biotyper (Bruker, Daltonics, Bremen, Germany) was used to identify microorganisms isolated from deer meat samples based on the comparison of the obtained patters with reference libraries.

### 2.4. Preparation of MALDI Matrix Solution

A stock solution that served as an organic reagent was prepared including: 50% of acetonitrile, 47.5% of water and 2.5% of trifluoroacetic acid (1 mL of stock solution was the mixture of 500 μL of pure acetonitrile, 475 μL of distilled water and 25 μL of pure trifluoroacetic acid). Two hundred and fifty microliter of the organic solvent was prepared and mixed in an Eppendorf flask with “HCCA matrix portioned”. All chemicals for matrix preparation were purchased from Lambda Life (Bratislava, Slovakia).

### 2.5. Sample Preparation and Identification

Samples were prepared as previously described [22]. Briefly, eight colonies per plate were analysed. The biological material was added from a Petri dish to an Eppendorf flask with 300 μL of distilled water, mixed and further 900 μL of ethanol was added. The mixture was then centrifuged at 10,000× *g* for 2 min (ROTOFIX 32A, Ites, Vranov, Slovakia). After discarding the supernatant, the precipitate was allowed to dry at 20 °C. Then 30 μL of 70% formic acid and 30 μL of acetonitrile were added to the pellet. Subsequently, the mixture was centrifuged at 10,000× *g* for 2 min. The supernatant of 1 μL was pipetted onto a MALDI plate, left to dry, and immediately, 1 μL of MALDI matrix solution was pipetted onto the plate. After drying, the samples were prepared for identification of microorganisms in a MALDI-TOF mass spectrometer (Bruker, Daltonics, Bremen, Germany). Mass spectra were automatically generated using the microflex LT MALDI-TOF mass spectrometer (Bruker Daltonics, Bremen, Germany) operated in the linear positive mode within a mass range of 2000–20,000 Da. The instrument was calibrated using the Bruker bacterial test standard. Results of mass spectra were processed with the MALDI Biotyper 3.0 software (Bruker Daltonics, Bremen, Germany). The identification criteria used were a score of 2.300 to 3.000 indicated highly probable identification on species level; a score of 2.000 to 2.299 secure genus identification with probable species identification; a score of 1.700 to 1.999 probable identification to the genus level; <1.700 was considered as unreliable identification.

### 2.6. Statistical Analysis

All measurements and analyses were carried out in triplicate. Statistical analysis and means comparison were run using software SPSS 26.0 (IBM SPSS Statistics, Chicago, IL, USA). ANOVA test was used to test the differences. Tukey HSD test was used for means comparison (95% confidence level).

## 3. Results

The essential oil from *Litsea cubeba* Pers. Fruit. was composed of 39.39% geranial, 29.49% neral, 14.29% α-limonene, 2.29% β-pinene, 1.89% sabinene, 1.89% 1,8-cineole, 1.69% α-pinene, 1.59% α-terpinolene and 1.32% 6-methyl-5-hepten-2-one, as major compounds adding up to 96% of the total profile [21].

### 3.1. Microbial Counts

Packaging conditions and storage time affected total viable counts (TVC) of deer meat (Table 1). Average initial counts were 2.00 ± 0.01 log CFU/g and after 20 days of refrigerated storage, they ranged from 5.17 log CFU/g on samples with 1% *Litsea cubeba* EO to the highest counts, 5.50 log CFU/g for the control sample. The average numbers of TVC were significantly lower (*p* < 0.05) in the samples treated with *Litsea cubeba* EO in comparison with control groups throughout the storage period, the highest differences were detected at 15 days of refrigerated storage (Table 1).

Coliforms bacteria (CB) counts are presented in Table 2. Initial counts of coliforms were of one logarithmic unit and not detected after one day of storage in all meat batches. Storage and packaging conditions significantly (*p* < 0.05) affected CB counts. Vacuum packaging itself slightly reduced CB counts as compared to aerobically stored meat. The presence of rapeseed oil proved better inhibitory effect than vacuum itself; the addition of 0.5% EO from *Litsea cubeba* slightly enhanced the preservation effect as compared to rapeseed oil addition. The presence of 1.0% *Litsea cubeba* EO completely inhibited CB during 20 days of refrigerated storage. 

Bacteria of the genus *Pseudomonas* were detected only in air packaged control group and vacuum packaged control group. The highest number of *Pseudomonas* was 1.96 ± 0.02 log CFU/g in the aerobic control group and 1.50 ± 0.01 log CFU/g in the control group with vacuum packaging on the twentieth day of storage The average numbers of *Pseudomonas* were significantly higher (*p* < 0.05) in the aerobic control group in comparison with control group with vacuum packaging. *Pseudomonas* was not detected in the control with rapeseed oil, nor in samples treated with 0.5 and 1.0% essential oil from *Litsea cubeba* (Table 3).

Counts of lactic acid bacteria (LAB) (Table 4) were kept low and were quite similar among batches (packaging conditions), they were mainly affected by storage time (an increase of 1 log unit after 20 days of storage). Significant differences were detected between batches, although in a very narrow range (mainly under 0.2 log units). 

### 3.2. Identification of Isolated Microorganisms

Microbial species were isolated from individual samples (Table 5). *Bacillus cereus* was isolated from all groups of samples, *Staphylococcus epidermidis* and *Staphylococcus capitis* were isolated from all control groups of samples and *S. capitis* was isolated from samples treated with 0.5% *Litsea cubeba* EO. Four species of the genus *Pseudomonas* (*Pseudomonas lundensis*, *Pseudomonas fragi*, *Pseudomonas taetrolens* and *Pseudomonas corrugata*) were isolated from control groups of samples. *Sphingomonas paucimobilis* and *Sphingomonas leidyi* were isolated from the aerobic control group and from control group with vacuum packaging. *S. leidyi* was isolated from control group with rapeseed oil and from samples treated with 0.5% *Litsea cubeba* EO. *Brevibacillus borstelensis* was isolated from the aerobic control group and *Pantoea agglomerans* from control group with vacuum packaging. The classification of microorganisms into families is shown in Table 6.

The most commonly isolated bacteria belong to the family Pseudomonadaceae (46%). The second most represented family is Staphylococcaceae (19%). The other families represented were Sphingomonadaceae (16%), Bacillaceae (14%), Paenibacillaceae (3%) and Enterobacteriaceae (2%) (Figure 1).

## 4. Discussion

Essential oil can be obtained from several parts of *Litsea cubeba*, the best antimicrobial activity is provided by fruit oil [18]. Predominant ingredient of *Litsea cubeba* fruit EO citral is known to possess a broad-spectrum antimicrobial activity against various bacteria and fungi. *Litsea cubeba* EO has a very unique sensory profile [23,24] which makes it suitable to applications in the preservation of a wide variety of foods [25]. In the present study, citral (geranial + neral) accounted for almost 70% of the composition of the essential oil, similar to that reported by Yang et al. [23]. They reported that LCEO was highly effective against Gram-negative bacteria and proposed a mechanism of action based on damages to the cell membrane and wall. Moreover, good inhibitory effectiveness against Gram-positive bacteria has been reported from a LCEO with similar composition [18,21,26], some studies even point to a slight higher sensitivity of Gram-positive bacteria to citral [27]. Conversely, other authors, testing a LC fruit EO containing 85% citral reported that Gram-positive bacteria were far less sensitive to LCEO than Gram-negative, although in both cases it was strongly dependent on the strain [19]. Other authors reported different composition of the fruit oil, such as 63.75% neral and 7.38% limonene as major compounds, and also showed excellent antimicrobial activity against Gram-positive bacteria [17]. In the present study, higher effectivity was observed against Gram-negative bacteria (coliforms and pseudomonas), whereas the effect against Gram-positive was much lower (LAB). The hydrophilic nature of citral (aldehyde) allows its adsorption to the bacterial surface and may disrupt cell integrity; aldehydes may also interact with functional groups of proteins, hence enhancing the inhibitory activity against bacteria. Several studies have evaluated antimicrobial mechanisms of action of LCEO and proved cell damages by detecting leaking of alkaline phosphatase and nucleic acids, enhanced electrical conductivity of the media, as well as microscopic examination [19,23]. Even if most studies in the scientific literature report successful in vitro antimicrobial activity of LCEO, the oils have limited miscibility with water and hence limited application in foods. Further technological developments in the application of this EO to foods, such as nano-emulsification, showed enhanced solubility and effectiveness of LCEO as antimicrobial, antioxidant and anti-biofilm agent [24].

When dealing with game meat, scarce data are available on their microbial quality. Peruzy et al. [28] analysed the initial bacterial contamination of fresh wild boar meat and reported counts far higher than the ones in the present study. They found the average TVC 4.76 log CFU/g, average Lactobacillus spp. 3.65 log CFU/g, average counts of *Pseudomonas* spp. 3.82 log CFU/g. Borilová et al. [29] analysed wild boar meat stored under aerobic conditions at 0 °C for a period of 21 days. *Lactobacillus* spp. counts were 3 log CFU/g and TVC counts were 2 log CFU/g at the end of the experiment, such counts are closer to the observed in the present study, being TVC much lower than the present deer samples, probably explained by the low storage temperature of 0 °C of the wild boar meat.

In the present study, TVC increased 3–3.5 log CFU/g during 20 days of refrigerated storage almost for all treatments, reaching values between 5 and 5.5 log CFU/g. Other authors reported an increase of TVC by 1 log CFU/g in vacuum-packaged deer meat after 7 days of storage [30], which is in agreement with the present results at 5–10 days of storage. Whereas in vacuum-packaged game meat, increases of 2 an 4 log CFU/g were reported after 7 and 12 days of storage, respectively [31], which are higher to the present increases. Very little information is available on the application of EO for the preservation of game meat. When oregano EO was applied to wildebeest meat samples stored under aerobic conditions, counts after 12 days of storage were 7 log CFU/g, the same TVC of control samples on the seventh day of storage [32]. In the same study, counts of Lactobacillus spp. in wild boar meat samples treated with 1% oregano EO and stored at 2.5 °C were of 7 log CFU/g on the thirteenth day of refrigerated storage, far higher than LAB counts observed in deer meat treated with LCEO and stored under anaerobic conditions in the present study. Initial counts were higher than the present. They reported that during aerobic storage of meat for 12 days, there was an average increase in the number of bacteria (*Lactobacillus* spp., CB, TVC) by 2 log CFU/g, The study concluded that the application of oregano EO to game meat can reduce the number of TVC, coliforms and LAB by about 1.4 times as compared to untreated aerobically packaged meat, and, finally could extend its shelf life by 3 days [32]. In the present study the effectiveness of LCEO against TVC and LAB was not as successful as the reported for oregano, also the storage temperature was higher in our study.

Meat from wild hunted animals may contain high microbial load, such contamination can be caused by poor hygiene of raw meat, caused by faecal contamination, either by unprofessional shooting or improper handling of the animal’s body [33,34], this is the reason of the large differences among the microbial load reported in the scientific literature. Avagnina et al. [35] compared the values of coliforms bacteria in wild boar meat samples with deer meat samples. They reported higher numbers of coliforms bacteria in wild boar meat (3.0 log CFU/g) than in deer meat (1.8 log CFU/g); in the present study even lower counts were detected at day zero (Table 2), and after 20 days of storage, coliforms were completely inhibited by 1% LCEO and reached up to 3 log CFU/g in control deer meat. As discussed previously, several authors reported good in vitro inhibition of Enterobacteria and, coliforms such as *Escherihia coli* with low (0.125%) concentration of LCEO [36]. As for *Pseudomonas*, they were only detected in 20 days stored control aerobic and vacuum stored samples, LCEO effectively inhibited their growth (Table 3 and Table 5), and although rapeseed oil reduced their counts they were not completely inhibited as they could be identified from rapeseed treated samples (Table 5).

Limited information is available on the microflora of game meat. Regarding wild boar meat, the most frequently isolated families are *Pseudomonas* (77%), *Pantoea* (73%), *Escherichia* (59%), *Acinetobacter* (55%), and also a high incidence of *Salmonella* (32%) (2019) [28]. Asakura et al. [37] analysed venison meat samples and they isolated the Shiga toxin producing *Escherichia coli* serotype from deer meat. They also identified coliforms bacteria *E. coli* and bacteria of the genus *Acinetobacter* and *Arthrobacter* from wild boar meat samples. Maksimovic et al. [33] isolated coliforms bacteria as well as a high number of *Bacillus cereus* bacteria from deer sausage samples. The species isolated from game meat in the present study have some similarities: B. cereus and the relevance of *Pseudomonas*; however the detected profile is quite different, and it is free of coliforms showing a higher hygienic quality of the meat. As can be seen in Table 5, vacuum, oil, and LCEO selectively inhibited different species both Gram-negative and Gram-positive, however most species isolated from LCEO treated samples were Gram-positive, so reinforcing the theory of a higher effectiveness of LCEO against Gram-negative bacteria.

## 5. Conclusions

Our results show that 0.5% and 1.0% *Litsea cubeba* EO applied to game meat in combination with vacuum packaging is highly effective against bacteria from genus *Pseudomonas* and coliforms. The use of 1.0% LCEO is moderately effective against total viable counts and lactic acid bacteria. The presence of LCEO reduces microbial load and diversity being more effective inhibiting Gram-negative and leaving a microbial ecosystem with prevalence of *Bacillus* and *Pseudomonas*. Inactivation of microorganisms in foods enhance food safety and has a positive effect on extending of their shelf life. LCEO, a natural and mild-flavour antimicrobial is suitable for extending the shelf life of vacuum-packaged deer meat. Studies are needed to further enhance the inhibition of total viable counts.

## Figures and Tables

**Figure 1 animals-12-02315-f001:**
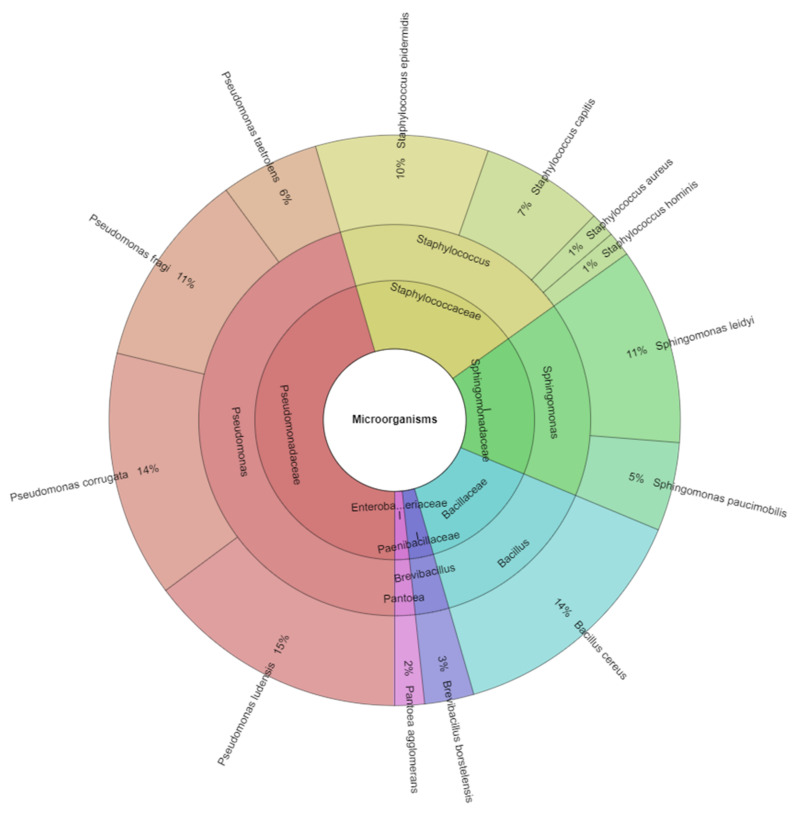
Identified species and family of bacteria in the deer meat.

**Table 1 animals-12-02315-t001:** Average numbers of total viable counts (TVC) log CFU/g in samples of deer meat during 20 days of storage at 4 °C.

Sample	TVC (log CFU/g)					
	0. Day	1 Day	5 Day	10 Day	15 Day	20 Day
Control-air	2.00 ± 0.01	2.20 ± 0.01 ^a^	3.38 ± 0.03 ^a^	4.12 ± 0.02 ^a^	4.73 ± 0.02 ^a^	5.50 ± 0.00 ^a^
Control-vacuum	2.00 ± 0.01	2.15 ± 0.03 ^a^	3.31 ± 0.01 ^a^	4.14 ± 0.02 ^a^	4.70 ± 0.01 ^a^	5.42 ± 0.01 ^a^
Control-rapeseed oil	2.00 ± 0.01	2.19 ± 0.00 ^a^	3.32 ± 0.02 ^a^	4.14 ± 0.01 ^a^	4.67 ± 0.02 ^a^	5.39 ± 0.01 ^a^
*Litsea cubeba* EO 0.5	2.00 ± 0.01	1.98 ± 0.01 ^b^	3.04 ± 0.03 ^b^	3.80 ± 0.01 ^b^	4.01 ± 0.01 ^b^	5.21 ± 0.02 ^b^
*Litsea cubeba* EO 1	2.00 ± 0.01	2.03 ± 0.05 ^b^	3.06 ± 0.01 ^b^	3.83 ± 0.02 ^b^	4.10 ± 0.01 ^b^	5.17 ± 0.02 ^b^

Control-air—aerobically packaged control samples; Control-vacuum—vacuum packaged control samples; Control vegetable oil—vacuum packaged control samples treated with rapeseed oil; *Litsea cubeba* EO 0.5—vacuum packaged samples treated with 0.5% *Litsea cubeba* EO; *Litsea cubeba* EO 1—vacuum packaged samples treated with 1.0% *Litsea cubeba* EO. ^a,b^ Different letters within the same column denote significant differences (*p* < 0.05).

**Table 2 animals-12-02315-t002:** Average numbers of coliforms bacteria (CB) log CFU/g in samples of deer meat during 20 days of storage at 4 °C.

Sample	Coliforms Bacteria (log CFU/g)					
	0 Day	1 Day	5 Day	10 Day	15 Day	20 Day
Control-air	1.02 ± 0.02	0.00 ± 0.00	1.52 ± 0.02 ^a^	2.06 ± 0.01 ^a^	2.29 ± 0.00 ^a^	3.00 ± 0.01 ^a^
Control-vacuum	1.02 ± 0.02	0.00 ± 0.00	1.11 ± 0.01 ^b^	1.20 ± 0.01 ^b^	1.21 ± 0.01 ^b^	2.47 ± 0.02 ^b^
Control-rapeseed oil	1.02 ± 0.02	0.00 ± 0.00	0.00 ± 0.00 ^c^	0.00 ± 0.00 ^c^	1.01 ± 0.00 ^c^	1.33 ± 0.03 ^c^
*Litsea cubeba* EO 0.5	1.02 ± 0.02	0.00 ± 0.00	0.00 ± 0.00 ^c^	0.00 ± 0.00 ^c^	1.07 ± 0.02 ^c^	1.21 ± 0.01 ^d^
*Litsea cubeba* EO 1	1.02 ± 0.02	0.00 ± 0.00	0.00 ± 0.00 ^c^	0.00 ± 0.00 ^c^	0.00 ± 0.00 ^d^	0.00 ± 0.00 ^e^

Control-air—aerobically packaged control samples; control-vacuum—vacuum packaged control samples; control vegetable oil—vacuum packaged control samples treated with rapeseed oil; *Litsea cubeba* EO 0.5—vacuum packaged samples treated with 0.5% *Litsea cubeba* EO; *Litsea cubeba* EO 1—vacuum packaged samples treated with 1.0% *Litsea cubeba* EO. ^a,b,c^ Different letters within the same column denote significant differences (*p* < 0.05).

**Table 3 animals-12-02315-t003:** Average numbers of *Pseudomonas* spp. log CFUg in samples of deer meat during 7 days of storage at 4 °C.

Sample	*Pseudomonas* (log CFU/g)					
	0 Day	1 Day	5 Day	10 Day	15 Day	20 Day
Control-air	0.00 ± 0.00	0.00 ± 0.00	1.09 ± 0.01 ^a^	1.25 ± 0.02 ^a^	1.47 ± 0.02 ^a^	1.96 ± 0.02 ^a^
Control-vacuum	0.00 ± 0.00	0.00 ± 0.00	0.00 ± 0.00 ^b^	1.12 ± 0.01 ^a^	1.31 ± 0.01 ^b^	1.50 ± 0.00 ^b^
Control-rapeseed oil	0.00 ± 0.00	0.00 ± 0.00	0.00 ± 0.00 ^b^	0.00 ± 0.00 ^c^	0.00 ± 0.00 ^c^	0.00 ± 0.00 ^c^
*Litsea cubeba* EO 0.5	0.00 ± 0.00	0.00 ± 0.00	0.00 ± 0.00 ^b^	0.00 ± 0.00 ^c^	0.00 ± 0.00 ^c^	0.00 ± 0.00 ^c^
*Litsea cubeba* EO 1	0.00 ± 0.00	0.00 ± 0.00	0.00 ± 0.00 ^b^	0.00 ± 0.00 ^c^	0.00 ± 0.00 ^c^	0.00 ± 0.00 ^c^

Control-air—aerobically packaged control samples; Control-vacuum—vacuum packaged control samples; Control vegetable oil—vacuum packaged control samples treated with rapeseed oil; *Litsea cubeba* EO 0.5—vacuum packaged samples treated with 0.5% *Litsea cubeba* EO; *Litsea cubeba* EO 1—vacuum packaged samples treated with 1.0% *Litsea cubeba* EO. ^a,b,c^ Different letters within the same column denote significant differences (*p* < 0.05).

**Table 4 animals-12-02315-t004:** Average numbers of lactic acid bacteria (LAB) log CFU/g in samples of deer meat during 20 days of storage at 4 °C.

Sample	LAB (log CFU/g)					
	0 Day	1 Day	5 Day	10 Day	15 Day	20 Day
Control-air	1.11 ± 0.02	1.30 ± 0.02 ^c^	1.57 ± 0.02 ^c^	1.60 ± 0.00 ^d^	1.78 ± 0.00 ^b^	2.06 ± 0.01 ^c^
Control-vacuum	1.11 ± 0.02	1.48 ± 0.03 ^a^	1.78 ± 0.00 ^a^	1.82 ± 0.02 ^a^	1.99 ± 0.01 ^a^	2.13 ± 0.02 ^b^
Control-rapeseed oil	1.11 ± 0.02	1.49 ± 0.01 ^a^	1.64 ± 0.03 ^b^	1.71 ± 0.00 ^b^	1.98 ± 0.01 ^a^	2.18 ± 0.01 ^a^
*Litsea cubeba* EO 0.5	1.11 ± 0.02	1.08 ± 0.02 ^d^	1.57 ± 0.02 ^c^	1.65 ± 0.00 ^c^	1.80 ± 0.02 ^b^	2.10 ± 0.01 ^b^
*Litsea cubeba* EO 1	1.11 ± 0.02	1.38 ± 0.06 ^b^	1.51 ± 0.02 ^d^	1.63 ± 0.02 ^c^	1.71 ± 0.01 ^c^	2.01 ± 0.02 ^d^

Control-air—aerobically packaged control samples; Control-vacuum—vacuum packaged control samples; Control vegetable oil—vacuum packaged control samples treated with rapeseed oil; *Litsea cubeba* EO 0.5—vacuum packaged samples treated with 0.5% *Litsea cubeba* EO; *Litsea cubeba* EO 1—vacuum packaged samples treated with 1.0% *Litsea cubeba* EO. ^a,b,c,d^ Different letters within the same column denote significant differences (*p* < 0.05).

**Table 5 animals-12-02315-t005:** Isolated bacteria from samples of deer meat stored under different packaging conditions.

	*Bacillus*	*Staphylococcus*	*Pseudomonas*	*Sphingomonas*	*Brevibacillus*	*Pantoea*
C-air	*B.cereus*	*S. epidermidis* *S. aureus* *S. hominis*	*P. lundensis* *P. fragi* *P. taetrolens* *P. corrugata*	*S. paucimobilis* *S. leidyi*	*B. borstelensis*	
CV	*B.cereus*	*S. epidermidis* *S. capitis*	*P. lundensis* *P. fragi* *P. taetrolens*	*S. leidyi* *S. paucimobilis*		*P. agglomerans*
CO	*B. cereus*	*S. epidermidis* *S. capitis*	*P. lundensis* *P. taetrolens*	*S. leidyi*		
LCEO 0.5	*B. cereus*	*S. capitis*		*S. leidyi*		
LCEO 1.0	*B. cereus*					

C-air—aerobically packaged control samples; CV—vacuum-packaged control samples; CO—vacuum packaged control samples treated with rapeseed oil; LCEO 0.5—vacuum-packaged samples treated with 0.5% *Litsea cubeba* EO; LCEO 1.0—vacuum-packaged samples treated with 1.0% *Litsea cubeba* EO.

**Table 6 animals-12-02315-t006:** Families of isolated microorganisms.

Microorganisms	Family
*Pseudomonas lundensis*, *Pseudomonas fragi*	Pseudomonadaceae
*Pseudomonas taetrolens*, *Pseudomonas corrugata*	
*Staphylococcus capitis*, *Staphylococcus epidermidis*	Staphylococcaceae
*Staphylococcus aureus*, *Staphylococcus hominis*	
*Sphingomonas leidyi*	Sphingomonadaceae
*Sphingomonas paucimobilis*	
*Bacillus cereus*	Bacillaceae
*Pantoea agglomerans*	Enterobacteriaceae
*Brevibacillus borstelensis*	Paenibacillaceae

## Data Availability

The data presented in this study are available on request from the corresponding author.

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
