# Peer review of "Microbiological Quality of Deer Meat Treated with Essential Oil Litsea cubeba"

_animals, 2022, doi:10.3390/ani12182315_

Round 1
Reviewer 1 Report
in the scope of the entire work - correct trimming %
for tables, in the notes, indicate the Latin names of microoganismd, as in the tables
indicate °C correctly in the scope of the entire work
Author Response
Reviewer #1
The Authors are very grateful to the Reviewer for valuable comments. We would like to thank the Reviewer for the time devoted for constructive and important comments to improve our paper.
Point 1: in the scope of the entire work - correct trimming %
Response: Trimming % has been corrected in entire work.
Point 2: for tables, in the notes, indicate the Latin names of microorganisms, as in the tables
Response: Names have been corrected as indicated
Point 3: indicate °C correctly in the scope of the entire work
Response: It has been corrected.
Reviewer 2 Report
Publication on game meat safety is very rare and scarce. The reviewer welcomes the idea of the authors. However, an extensive review of the paper for the flow and grammar should be done before its publication.
Ln 34: Different types of packaging are most often used to extend the shelf life… Reviewers recommend the author to be clear on what they are trying to say or write a complete sentence.
Ln 36: Replace ‘They can be…’ with ‘Packaging can be…’
Ln 38: Replace ‘There were…’ with ‘Vacuum packaged deer meat was evaluated for the microbiological quality…’
Suggestion:
A simple summary should be re-written, and the reviewer encourages authors to make it very simple and clear.
Ln 44: During storage of meat for how many days?
Ln 45: Italicize Pseudomonas
Ln 48: Delete ‘without treatment’
Ln 50: During the entire storage period? If so, replace ‘twenty days’ ‘entire’
Ln 51: Delete ‘without treatment’. Control groups are either positive or negative controls. Are there treated meat products without vacuum packaging? This should be clear in the abstract section.
Ln 52: Delete ‘without treatment’
Ln 54: …frequently isolated bacteria from deer meat were…
Ln 64: …A, C, E AND B, as well as minerals…
This reviewer did not see connections between the paragraphs and lost the flow. Well-connected paragraphs keep readers focused, and thus understand the content.
Ln 70: What is the author trying to convey by stating ‘regional/traditional diet’?
Ln 71-74: This line should be re-written – For example: antemortem should come first before hunting.
Ln 75: The reviewer recommends that the author write a separate paragraph for microbiological quality, which should be combined with the fourth paragraph of your introduction.
Ln 79: Pseudomonas spp. is dominant in spoilage meat.
Ln 80: vacuum -packed or packaged?
Ln 92: …essential oils (EO) that help in defending …
Ln 112: The sentence is not clear
Ln 113: This biotyper has several …
Ln 118: …quality of vacuum packaged deer meat stored…for 20 days and treated with …
In the materials and method section, ‘Thigh muscle’ isn’t enough for the description. Which thigh muscle did the authors use in this research? For example- Sartorius, Tensor fasciae latae, etc.
How did the authors estimate the deer’s age? Of course, there are some methods to find animals’ age.
What was the condition of the meat before purchasing? Was the meat chopped, diced, or ground? Any idea on how long it was there in the store?
Was there any reason or citation behind using only 5g of meat sample?
What is the experimental design? How many samples per treatment and replication?
A total of 120 samples for one replication or three replications?
Is cultivation similar to plating techniques? If not, the authors must explain the technique more to make it clear.
Ln 257-268: Italicize all scientific names
Author Response
Reviewer #2
Publication on game meat safety is very rare and scarce. The reviewer welcomes the idea of the authors. However, an extensive review of the paper for the flow and grammar should be done before its publication.
The Authors are very grateful to the Reviewer for valuable comments. We would like to thank the Reviewer for the time devoted for constructive and important comments to improve our paper.
Summary section:
Point 1: Ln 34: Different types of packaging are most often used to extend the shelf life… Reviewers recommend the author to be clear on what they are trying to say or write a complete sentence.
Response: The sentence has been rewritten to enhance clarity types of packaging which can extend the shelf life of meat are listed in the line 35.
Point 2: Ln 36: Replace ‘They can be…’ with ‘Packaging can be…’
Response: It has been replaced as suggested.
Point 3: Ln 38: Replace ‘There were…’ with ‘Vacuum packaged deer meat was evaluated for the microbiological quality…’
Response: It has been replaced as suggested.
Point 4: Suggestion: A simple summary should be re-written, and the reviewer encourages authors to make it very simple and clear.
Response: The summary has been rewritten to enhance clarity.
Point 4: Ln 44: During storage of meat for how many days?
Response: The storage time of meat was added (20 days)
Point 5: Ln 45: Italicize Pseudomonas
Response: It has been corrected
Point 6: Ln 48: Delete ‘without treatment’
Response: ‘without treatment’ has been changed to ‘aerobically packaged control group‘ throughout the manuscript
Point 7: Ln 50: During the entire storage period? If so, replace ‘twenty days’ ‘entire’
Response: It has been replaced as suggested.
Point 8: Ln 51: Delete ‘without treatment’. Control groups are either positive or negative controls. Are there treated meat products without vacuum packaging? This should be clear in the abstract section.
Response: The term ‘without treatment’ has been changed to ‘aerobically packed control group‘. There are three controls: air, vacuum, and vacuum with rapeseed oil (not containing essential oil). Packaging of deer meat samples are explained in the Material and methods section – lines 135-148.
Point 9: Ln 52: Delete ‘without treatment’
Response: Done as suggested
Point 10: Ln 54: …frequently isolated bacteria from deer meat were…
Response: It has been corrected as suggested
Point 11: Ln 64: …A, C, E AND B, as well as minerals…
Response: It has been corrected
Point 12: This reviewer did not see connections between the paragraphs and lost the flow. Well-connected paragraphs keep readers focused, and thus understand the content.
Response: The Introduction section has been reorganized and paragraphs connected to provide a better reading flow.
Point 13: Ln 70: What is the author trying to convey by stating ‘regional/traditional diet’?
Response: It is meant that consumers identify meat from wild animals as linked to traditional diets given that not all cultures, regions use wild meat for consumption.
Point 14: Ln 71-74: This line should be re-written – For example: antemortem should come first before hunting.
Response: This sentence has been rewritten to enhance clarity.
Point 15: Ln 75: The reviewer recommends that the author write a separate paragraph for microbiological quality, which should be combined with the fourth paragraph of your introduction.
Response: The introduction section has been reorganized as suggested by the reviewer.
Point 16: Ln 79: Pseudomonas spp. is dominant in spoilage meat.
Response: It has been corrected in the text
Point 17: Ln 80: vacuum -packed or packaged?
Response: It has been corrected – vacuum packaged, the term packaged is now used in the manuscript
Point 18: Ln 92: …essential oils (EO) that help in defending …
Response: It has been corrected
Point 19: Ln 112: The sentence is not clear.
Response: It has been rewritten to enhance clarity
Point 20: Ln 113: This biotyper has several …
Response: It has been corrected
Point 21: Ln 118: …quality of vacuum packaged deer meat stored…for 20 days and treated with …
Response: It has been corrected as suggested
Point 22: In the materials and method section, ‘Thigh muscle’ isn’t enough for the description. Which thigh muscle did the authors use in this research? For example- Sartorius, Tensor fasciae latae, etc. How did the authors estimate the deer’s age? Of course, there are some methods to find animals’ age. What was the condition of the meat before purchasing? Was the meat chopped, diced, or ground? Any idea on how long it was there in the store?
Response: More details are provided in the present version of the manuscript, deer meat was purchased from an authorized dealer, and we used only information on the packaging of meat (age, date of hunting…) so we added more information, the muscle (musculus semimebranosus), and the time after catch (48 hour after the catch).
Point 23: Was there any reason or citation behind using only 5g of meat sample?
Response: The amount of 5 g of sample in 45 mL of saline solution is commonly used as basic dilution in microbiological analysis and allowed minimized handling of samples.
Point 24: What is the experimental design? How many samples per treatment and replication?
Response: Anova tests were used to evaluate the effect of the type of packaging and storage time on microbial counts. A total 120 samples were used in the experiment. All measurements and analyses were carried out in triplicate. The number of samples and replications are given in the section Materials and methods.
Point 25: A total of 120 samples for one replication or three replications?
Response: A total 120 samples were used for three replications.
Point 26: Is cultivation similar to plating techniques? If not, the authors must explain the technique more to make it clear.
Response: This section has now been rewritten to enhance clarity; the term cultivation has been avoided.
Point 27: Ln 257-268: Italicize all scientific names
Response: It has been corrected
Reviewer 3 Report
Please see the attached document outlining the suggested changes to the presentation of this material.

Author Response
Reviewer #3
Overall, the authors have done a great job expanding the existing knowledge of storing game meat for consumer uses. However, there are some changes outlined below that the authors should consider improving the presentation of the presented work. See below by line number the suggested changes to consider.
The Authors are very grateful to the Reviewer for valuable comments. We would like to thank the Reviewer for the time devoted for constructive and important comments to improve our paper.
Point 1: Line 4: Some of the authors have multiple designations. Confirm that multiple institutions should be represented with author designees.
Response: We confirm that some authors have multiple designations.
Point 2: Line 45: Replace zeroth with day 0, 1, 5, 10, 15 and 20.
Response: It has been replaced as suggested
Point 3: Line 53: Replace twentieth day with day 20.
Response: It has been replaced as suggested
Point 4: Keywords: would recommend reconsidering the use of game meat or venison as key words.
Response: Game meat was removed from Key words
Point 5: Line 65: Deer meat is richer in calcium….. than beef? Likely needs a reference to support this statement.
Response: This sentence has been rewritten in the present version avoiding comparisons.
Point 6: Line 111: Wild meat or game meat? Consistency in presentation to avoid reader confusion.
Response: It has been unified in the article, now game meat is used.
Point 7: Line 119: Presentation of 4 °C. Presented throughout text with a space, but on this line there is no space. Correct for consistency.
Response: It has been corrected.
Point 8: Line 124 – 127: Provide an estimated age of meat at the time of purchase. How long from harvest to store presentation and then use within the study. Did this timeframe occur after harvest and processing after 12 hours, 24 hours, 48 hours? Age of raw material can have an inherent impact on a study measuring microbial growth over time.
Response: More details are provided in the present version of the manuscript, deer meat was purchased from an authorized dealer, and we used only information on the packaging of meat (age, date of hunting…) so we added more information, the muscle (musculus semimebranosus), and the time after catch (48 hour after the catch).
Point 9: Line 128: Describe hygienic conditions? Transported under refrigerated conditions to the storage location?
Response: The conditions of transport have been added
Point 10: Line 138: The authors do not mention the use of rapeseed oil either in the title, keywords, or abstract. It appears that this treatment could be confounding to the entire study design. Recommend considering removal of this treatment from the study and focus the presentation on the Control and Vacuum packaging with Litsea cubeba only. The authors have provided a clear description of the treatments, but it appears that the study flaw lies with treatment 3 (Control with rapeseed oil). Authors need to provide an explanation why Control with litsea was not included and replaced with rapeseed oil. What percentage of rapeseed oil was used?
Response: Rapeseed oil was used for the preparation of 0.5 and 1% Litsea cubeba essential oil solutions, which justifies the need to evaluate a vacuum packaged meat treated with rapeseed oil as a control. Now it has been better clarified in the manuscript.
Point 11: Line 138: Authors should provide the formulation for creating a 0.5% and 1.0% solution. Was this sprayed, dipped, or soaked for defined period? Did this solution include water? Was this amount applied to the meat at a percent of the green weight?
Response: Essential oil was dissolved in commercial food grade rapeseed oil. The way of application of the EO has been added (soaking for 30 min).
Point 12: Line 148: Replace the zeroth, fifth with: Microbiological analyses were performed on day 0, 5, 10, 15 and 20th. Is day 1 missing from this list?
Response: It has been corrected - that was a mistake - analysis were also performed on 1st day
Point 13: Line 150: What instruments were used to carry out homogenization? (Model, Manufacturer, City, Country).
Response: It has been added to the material and methods section.
Point 14: Line: Authors should include the supplier and manufacturer of chemicals used in creating the reagents for MALDI solution.
Response: It has been added to the material and methods section.
Point 15: Line 176: Model, manufacturer, City, Country for centrifuge.
Response: It has been added to the material and methods section.
Point 16: Line 177: Define room temperature in degrees Celsius.
Response: Temperature has been added
Point 17: Line 182: Model, manufacturer, City, Country for mass spectrometer. In addition, provide settings of the mass spec for analysing the sample.
Response: It has been added to the material and methods section.
Point 18: Line 187: Remove quotations around treatment and storage.
Response: Quotations have been removed
Point 19: Line 215: presentation of (p<0.05) should be consistent. Authors have presented previously with a lower case “p”. Correct for consistency.
Response: It has been corrected in the manuscript
Point 20: Line 257 – 268: Presentation of microbial species is inconsistent within the text. Authors should consider presentation in italics for consistency throughout the text.
Response: It has been corrected in entire text
Point 21: Line 281 – 284: Presentation of microbial species should be italicized? Authors should consider this presentation for clarity.
Response: Microbial families are not italicized, however all species name have been revised and italicized
Point 22: Line 315: Presentation of game meat vs. wild meat? Consistency in presentation.
Response: It was unified in the text as ‘game meat’
Point 23: Line 320: Presentation of degrees Celsius. Spacing of degree symbol and C.
Response: It has been corrected
Point 24: Line 326: What is roe deer meat? Should this be raw?
Response: It has been rewritten
Point 25: References: The presentation of references is lacking the doi: for each citation. Authors should include this information.
Response: DOI numbers have now been included.
Round 2
Reviewer 2 Report
Dear authors,
Thank you for providing the revised version of your manuscript. Here are few suggestions-
In your newly added paragraph -
Ln 87: Replace in deep with in-depth
Ln 88-89: This sentence is not clear - MALDI-TOF MS Biotyper is a fast and reliable method of identification that takes only a few minutes from cultivation the isolate.
Suggestion - MALDI-TOF MS Biotyper is a fast and reliable method that takes only a few minutes to identify microbes from the cultivation of an isolate.
Ln 89-90: This Biotyper has several unique features that make it one of the most successful methods of identification, and therefore that it is broad-spectrum, fast and sensitive [11]
Suggestion: This Biotyper has several unique features, making it one of the most successful identification methods [11].
Re-write this paragraph: Determining the microbial populations in meat would be a first step towards defining strategies for meat preservation, another contribution to meat safety would be testing preservation conditions such as packaging conditions and the use of natural antimicrobials. In this sense, plant extracts are natural compounds of interests for meat preservation.
Suggestion- Break it down into 2-3 sentences to clarify.
Ln 156: Were meat samples fully submerged in the essential oil?
Ln 166: Replace Coliform bacteria with Coliforms were
Ln 179: of 2.5% trifluoroacetic acid OR 2.5% of trifluoroacetic acid?
Author Response
The Authors are very grateful to the Reviewer for valuable comments. We would like to thank the Reviewer for the time devoted for constructive and important comments to improve our paper.
Thank you for providing the revised version of your manuscript. Here are few suggestions-In your newly added paragraph -
Point 1: Ln 87: Replace in deep with in-depth
Response: It was corrected.
Point 2: Ln 88-89: This sentence is not clear - MALDI-TOF MS Biotyper is a fast and reliable method of identification that takes only a few minutes from cultivation the isolate.
Response: It was corrected.
Point 3: Suggestion - MALDI-TOF MS Biotyper is a fast and reliable method that takes only a few minutes to identify microbes from the cultivation of an isolate.
Response: It was corrected.
Point 4: Ln 89-90: This Biotyper has several unique features that make it one of the most successful methods of identification, and therefore that it is broad-spectrum, fast and sensitive [11]
Suggestion: This Biotyper has several unique features, making it one of the most successful identification methods [11].
Response: It was corrected.
Point 5: Re-write this paragraph: Determining the microbial populations in meat would be a first step towards defining strategies for meat preservation, another contribution to meat safety would be testing preservation conditions such as packaging conditions and the use of natural antimicrobials. In this sense, plant extracts are natural compounds of interests for meat preservation.
Suggestion- Break it down into 2-3 sentences to clarify.
Response: It was re-described.
Point 6: Ln 156: Were meat samples fully submerged in the essential oil?
Response: The packaged samples were homogenized in the hands with the essential oil so that the EO was all over the surface of the meat.
Point 7: Ln 166: Replace Coliform bacteria with Coliforms were
Response: It was corrected.
Point 8: Ln 179: of 2.5% trifluoroacetic acid OR 2.5% of trifluoroacetic acid?
Response: 2.5% of trifluoroacetic acid. It was corrected.
Reviewer 3 Report
Thank you for providing the detail outlining corrections. Accept as presented.
Author Response
Reviewer #3
Thank you for providing the detail outlining corrections. Accept as presented.
We would like to thank the Reviewer for the time devoted for constructive and important comments to improve our paper.